# Evaluation of the Performance of a ZnO-Nanoparticle-Coated Hydrocolloid Patch in Wound Healing

**DOI:** 10.3390/polym14050919

**Published:** 2022-02-25

**Authors:** Van Anh Thi Le, Tung X. Trinh, Pham Ngoc Chien, Nguyen Ngan Giang, Xin-Rui Zhang, Sun-Young Nam, Chan-Yeong Heo

**Affiliations:** 1Department of Plastic and Reconstructive Surgery, Seoul National University Bundang Hospital, Seongnam 13620, Korea; vananhle2201@gmail.com (V.A.T.L.); xttrinh1199@gmail.com (T.X.T.); ngocchien1781@gmail.com (P.N.C.); nngiang1105@gmail.com (N.N.G.); zhangxinrui@snu.ac.kr (X.-R.Z.); 2Department of Plastic and Reconstructive Surgery, College of Medicine, Seoul National University, Seoul 03080, Korea

**Keywords:** wound healing, hydrocolloid, ZnO-NPs, inflammation, proliferation

## Abstract

Hydrocolloid dressings are an important method for accelerating wound healing. A combination of a hydrocolloid and nanoparticles (NPs), such as gold (Au), improves the wound healing rate, but Au-NPs are expensive and unable to block ultraviolet (UV) light. Herein, we combined zinc oxide nanoparticles (ZnO-NPs) with hydrocolloids for a less expensive and more effective UV-blocking treatment of wounds. Using Sprague–Dawley rat models, we showed that, during 10-day treatment, a hydrocolloid patch covered with ZnO-NPs (ZnO-NPs-HC) macroscopically and microscopically stimulated the wound healing rate and improved wound healing in the inflammation phase as shown by reducing of pro-inflammatory cytokines (CD68, IL-8, TNF-α, MCP-1, IL-6, IL-1β, and M1) up to 50%. The results from the in vitro models (RAW264.7 cells) also supported these in vivo results: ZnO-NPs-HCs improved wound healing in the inflammation phase by expressing a similar level of pro-inflammatory mediators (TNF-α and IL-6) as the negative control group. ZnO-NPs-HCs also encouraged the proliferation phase of the healing process, which was displayed by increasing expression of fibroblast biomarkers (α-SMA, TGF-β3, vimentin, collagen, and M2) up to 60%. This study provides a comprehensive analysis of wound healing by measuring the biomarkers in each phase and suggests a cheaper method for wound dressing.

## 1. Introduction

Wound healing has received attention worldwide [1,2,3,4]. Wound healing is classically divided into four stages: (A) hemostasis, (B) inflammation, (C) proliferation, and (D) remodeling [5]. Each stage is characterized by key molecular and cellular events and is coordinated by a host of secreted factors recognized and released by the cells that respond to the wound. In the inflammatory phase, neutrophils promote the formation of reactive oxygen species (ROS), and pro-inflammatory cytokines to clean the wound [6]. Next, neutrophils undergo apoptosis and are phagocytosed by the newly arrived macrophages. 

Macrophages also phagocytose bacteria and debris to clean the wound. The wound is disinfected and prepped for tissue regrowth during this time [7]. In the proliferative phase, wound cells (e.g., endothelial cells, fibroblasts, and keratinocytes) proliferate and migrate to regenerate the lost tissue. Granulation tissue (GT) (i.e., preliminary vascularized extracellular matrix-ECM), is then formed, and the wound is closed by keratinocytes. The ECM within the granulation tissue grows and gains mechanical strength during remodeling phase [8]. 

Apoptosis of myofibroblasts and vascular cells complete the wound healing process, leaving a collagen-rich scar behind. Numerous materials have been developed and renovated for a wound dressing that will provide a targeting reagent to facilitate the healing ability [9,10,11,12,13]. Traditional wound dressings include cotton bandage and gauze, but hydrogels, foams, tissue adhesives, vapor-permeable films, polyurethane (PU), alginates, and silicone meshes are among the products that form synthetic wound dressings [14,15,16,17,18]. 

However, the current materials are costly in terms of industrial production and their low biocompatibility and the subsequent severe environment prohibits cell growth, leading to slow tissue regeneration [19,20,21,22]. PU dressings are waterproof and bacteria resistant but are expensive, and their adhesion ability to the wound bed is low [23]. Alginate dressings include a calcium component that can be useful in blood clotting and acts as a hemostat [24]. However, alginate is not suited for dried wounds since it requires moisture to function properly [25]. Therefore, there is a need to innovate a novel material to address all these issues to promote wound healing.

A hydrocolloid (HC) comprises a gel-able self-adhesive impermeable layer implemented on a polyurethane film. HCs have been applied in wound treatment to absorb exudate leakage [26,27,28,29]. A hydrocolloid helps maintain a suitably moist environment for tissue development for injured areas without any interference of pathological agents [30,31]. Previous studies have shown that hydrocolloids decrease the healing time [25,32,33,34,35]. Therefore, hydrocolloids are suitable as base materials in wound dressings. However, a combination of a hydrocolloid and a compound capable of reducing inflammation and promoting wound healing has more potential.

Nanoparticles have garnered attention in the biomedical field as an innovative antibacterial material that can be applied in bio-imaging, tissue engineering, and in particular wound healing as nanoparticles liberate hydrogen peroxide supports the proliferation of fibroblasts [36,37,38]. The wound healing process is aided by zinc to increase collagen synthesis and fast re-epithelization [39]. 

Furthermore, zinc oxide nanoparticles (ZnO-NPs) have been used as an important ingredient in wound dressing products due to their ability to protect against UV light, their antibacterial properties, their capacity to increase swelling ratio of materials, and their interaction with cells that promote recovery from injury by accelerating cellular metabolism, DNA repair, cytokine regeneration, etc. [40,41,42,43,44,45,46,47,48]. Ågren et al. demonstrated a hydrocolloid dressing with ZnO-NPs that inhibited bacterial growth. Therefore, we chose ZnO-NPs as a potential candidate for integrating with a hydrocolloid to create a product that helps heal wounds [49].

Herein, we developed a novel hydrocolloid patch covered with ZnO-NPs (ZnO-NPs-HC) to promote wound healing in Sprague–Dawley rats. The patch exhibited effectiveness as a wound dressing, demonstrated by visualized immune experiments, such as immunofluorescence (IF) staining, Western blots, enzyme-linked immunosorbent assay (ELISA), and hematoxylin and eosin (H&E) staining. 

The sharp intensity of the Western blots clearly showed the release of essential cytokines (vimentin, TNF-α, IL-6, IL-8, etc.) to help recover from wounds in rat samples that had undergone hydrocolloid patch therapy. The IF staining assays proved the biocompatibility of the patch and simultaneously confirmed the rehabilitation of the injuries under the patch treatment. The other methods confirmed the effectiveness of ZnO-NPs-HCs in the inflammation and proliferation phases of the healing process.

## 2. Materials and Methods

### 2.1. Materials

The ZnO-NPs-HCs patch was kindly provided by CGBio company (Seoul, South Korea). The patch was prepared by following protocols in a pending patent.

Bovine Serum Albumin (BSA) and 10× Tris-Buffered Saline (TBS) were purchased from Biosesang company, Bundang, South Korea. Tween 20 (extra pure grade) was purchased from Duksan company, Ansan, South Korea. Fetal Bovine Serum (FBS) was purchased from Thermo Fisher, Waltham, MA, USA. 10× Phosphate-buffered saline (PBS) was purchased from iNtRON Biotechnology company, Seoul, South Korea. 1× Tris buffered saline with Tween (TBST) solution was made from 10× TBS and Tween 20.

### 2.2. In Vivo Experiment

A total of ten Sprague–Dawley rats (weighing 200–300 g) were used. The animals were almost 8 weeks old. The creatures were maintained in a 12-h/12-h light/dark cycle under specific-pathogen-free (SPF) conditions with free access to food and water. Bundang Seoul National University Hospital’s Institutional Animal Care and Use Committee authorized the experimental animal protocol (approval number: BA-1802-241-014-08). All procedures followed the NIH Guide for the Care and Use of Laboratory Animals. The rats were divided into two groups: control (NC, *n* = 5) and hydrocolloid (HC, *n* = 5) with the hydrocolloid dressing sample. 

Both groups were anesthetized using anesthetic wither and shaved on the dorsal by an electric hair clipper. The shaved area was disinfected with a povidone-iodine solution. A 10 mm diameter biopsy punch was used for making a full-thickness circular hole on the back of each rat. Hydrocolloid patches covered with ZnO-NPs were used to cover the wounds of the HC group before covering the wounds with commercially available film dressing (Tegaderm, 3 M, Saint Paul, MN, USA). All wounds were imaged at days 0, 3, 7, and 10 with a digital camera. 

After 10 days of treatment, the rats were asphyxiated with CO_2_, and 12 mm diameter circular hole with full-thickness skin around the wound samples were collected from each rat. Each skin sample was divided into two parts. The first part was fixed with 10% formaldehyde solution, dehydrated in a series of alcohol solutions (80–100%), embedded in paraffin, and cross-sectioned to obtain 5 µm thick slices for immunofluorescence (IF) staining and histological staining (e.g., hematoxylin-eosin, H&E, and Masson’s trichrome, MT). The second part was used for biochemistry analysis, such as Western blot [50], Enzyme-Linked Immunosorbent (ELISA) [51], DCF-DA [52], and nitrate assays [53].

### 2.3. Measurement of the Wound Healing Area

The wound area closure was computed by using the following equation:(1)$$\mathrm{Wound}\;\mathrm{closing}\;\mathrm{area}\;(\%)=\frac{A_{0}-A_{i}}{A_{0}}A\times 100\%$$
where A_0_ and A_i_ are the wound areas on day 0 and day i (0, 3, 7, and 10). A_0_ and A_i_ areas were obtained from microscopy images using ImageJ software [54].

### 2.4. Immunofluorescence Staining

The immunofluorescence staining protocol followed was according to the study of Zaquot and colleagues [55] with minor modifications. All tissues slices were deparaffinized using xylene, ethanol (100%, 95%, 90%, 80%, and 70%), and PBS. After that, for the heat antigen retrieval (unmasking) step, an antigen retrieval buffer (pH 6.0) containing 10% FBS in PBS and citrate buffer was used to microwave-boil all slices for 20 min. For the blocking step (about 1 h), 4% BSA in PBS was added to the slices. 

Next, for the permeabilization step, tissue slices were incubated with primary antibodies against α-SMA, vimentin, IL-1β, IL-6, IL-8, TNF-α, MCP-1, and CD68 (Santa Cruz Biotechnology; 1:100) overnight at 4 °C. The second antibody, Alexa fluor 488 goat anti-mouse IgG, was stained on the slices 1 h after three PBS washes. Finally, all tissues were mounted with 4′,6-diamidino-2-phenylindole (DAPI VECTASHIELD^®^) (Vector Laboratories, Burlingame, CA, USA) for fluorescence imaging and kept at −20 °C.

### 2.5. Protein Preparation

The wound tissues of the control and hydrocolloid groups on day 10 were homogenized using a protein extraction solution containing 1 mM PMSF and a protein inhibitor cocktail. The tissue extract was clarified by centrifugation at 14,000 rpm for 15 min at 40 °C, followed by collection of the supernatant. The protein solution was kept at −80 °C for later analysis by Western blot [50], ELISA [51], DCF-DA [52], and nitrite [53] assays.

### 2.6. Western Blot

We used Western blot analysis [50] to classify specific proteins from a complex mixture of proteins extracted from the tissue in the proliferation phase. Separation was accomplished by using sodium dodecyl sulfate–polyacrylamide gel electrophoresis, followed by protein blotting on a polyvinylidene fluoride membrane (Bio-Rad Laboratories, Hercules, CA, USA). 

After blocking with PBS containing 5% nonfat dry skim milk, the membranes were incubated with primary antibodies against β-actin, α-SMA, vimentin, TGF-β3 (Santa Cruz Biotechnology; 1:1000), collagen I (Abcam; 1:3260), and collagen III (Abcam; 1:5000) overnight at 4 °C by shaking the machine. The second incubation step used an (H + L)-HRP conjugate (Bio-Rad; 1:2000) and lasted 2 h. The signals were visualized using the ChemiDoc^TM^ Imaging System (Bio-Rad Laboratories, Hercules, CA, USA), and the band densities were quantified using ImageJ software [56,57,58].

### 2.7. ELISA

The ELISA analysis was conducted by following the protocol published elsewhere [51]. After coating a 96-well plate overnight at room temperature with 100 µL of capture antibodies against IL-6, IL-1β, IL-8, TNF-, and MCP-1 (Santa Cruz Biotechnology; 1:500), the coating solution was removed and washed twice with 200 μL of TBST. Next, the remaining protein-binding sites in the coated wells were blocked for 2 h by adding 200 µL of blocking buffer (10% FBS in PBS) to each well. After removing the blocking buffer, 100 μL of diluted samples was added to each well and incubated at 25 °C for 3 h. Then, 100 μL of a diluted detection antibody was added to each sample after washing the sample five times by using 200 μL of TBST. 

Next, the plate was washed five times with TBST before the immediate addition of 100 μL of a conjugated secondary antibody diluted in blocking buffer and incubated for 30 min at room temperature. Finally, 100 μL of the solution of the substrate reagent A and B (BD Biosciences) (ratio 1:1) was added to each sample for measuring the concentration of proteins based on the absorbance–concentration standard curve of albumin standards. Absorbance was measured at a 405 nm wavelength, and albumin standards were from the Pierce^TM^BCA protein assay kit (ThermoScientific, Lafayette, CO, USA).

### 2.8. Nitrite Assay

Nitric oxide regulates three essential aspects of the wound healing process, vascular homeostasis, inflammation, and antimicrobial action, particularly excessive NO associated with infected or highly inflamed wounds leading to tissue damage [53].

The Griess modified reagent was used to evaluate nitrite and NO oxidation products in wound lysates. The total protein was diluted with distilled water at a ratio of 1:1. A mix of 100 µL of the 1× Griess modified reagent and 100 µL of the diluted protein extract was incubated at room temperature for 15 min. A series of nitrite standard solutions of 0, 1.6125, 3.125, 6.25, 12.5, 25, and 50 µM were prepared for a standard curve. The absorbance was measured at the 504 nm wavelength.

### 2.9. DCF-DA

Protein samples, 100 µL in all, containing 20 µM of a DCF-DA solution, were incubated at 37 °C under 5% CO_2_ for 45 min in the dark. The fluorescence intensity was measured at 485 nm (excitation) and 535 nm (emission) using a microplate reader.

### 2.10. In Vitro Assays

RAW264.7 cells were cultured in DMEM media at 3.8 × 10^5^ cells/well in 24-wells plates (SPL Life Science) and incubated at 37 °C under 5% CO_2_ for 24 h. Then, the cells were activated with 1 µg/mL of LPS in media for three wells with hydrocolloid patches and three wells without hydrocolloid patches. After 24 h, cell fluid in media was collected for ELISA analysis.

### 2.11. Statistical Analysis

PRISM statistical software was used to depict the entire set of results as mean values ± SEM (GraphPad Software, San Diego, CA, USA). Unpaired Student’s *t*-test determines the significant differences between two groups’ means. For all analyses, the differences were considered significant.

## 3. Results

### 3.1. Macroscopic and Microscopic Observation of the Wound Healing Process

The macroscopic images in Figure 1A are of the wound on day 3, day 7, and day 10 after the wound was inflicted. After 10 days, the hydrocolloid-treated wound showed a much smaller wound area than that of the control group. Figure 1B shows the wound healing closure rates, expressed by the percentage of the wound area compared to the wound size on day 0. The percentages of the healing areas of the hydrocolloid-treated samples are 61%, 90%, and 98% on day 3, day 7, and day 10, respectively. The healing area percentages of the control group on the same days are 49%, 80%, and 85%, respectively.

H&E and MT staining were used to examine the microscopic wound alteration after hydrocolloid treatment (Figure 1C,D). On day 10, fresh epidermal and granulation tissue was taken from the wound locations and sectioned and examined under a microscope (Figure 1C). Statistical analysis of the H&E showed that the hydrocolloid-treated wounds produced a thicker epidermis and granulation tissues (48.2 ± 2.5 and 497.8 ± 9.7 μm) than the control (41.3 ± 1.8 and 420.3 ± 10.3 μm) (Figure 1E,F). 

The collagen deposition on the wound was also investigated by MT staining to further evaluate the normalization degree of collagen in injured skin (Figure 1D). Compared to the control wound, the hydrocolloid-treated wound had a considerably darker and more evenly distributed blue color, which means more collagen deposition (Figure 1D). Quantitative measurement revealed that the blue color density was 59.1 ± 1.9 for hydrocolloid-treated wounds, which was significantly higher than that of 50.3 ± 1.1 for the control (Figure 1G).

### 3.2. Observation on the Inflammation of the Wound Healing Process

While the level of NO in the control group is 25.2 ± 4.5, the level of NO in the treated group is only half of that (12.3 ± 0.7) (Figure 2A).

The results of reactive oxygen species (ROS) production measured by DCF-DA show that there was no significant difference between control and treated groups (Figure 2B). Pro-inflammatory cytokine expression was measured by using the ELISA kit to demonstrate the product’s capacity to reduce inflammation. On day 10, once the healing process was completed, the concentration of TNF-α in the hydrocolloid group (0.030 ± 0.003 ng/mL) was half that of the control group (0.060 ± 0.012 ng/mL) (Figure 2C). 

Furthermore, IL-6, MCP-1, and IL-8 all show a similar tendency and decline in varied ratios over the course of 10 days of therapy (Figure 2D–F). Immunofluorescence staining results provide qualitative and quantitative comparisons between the treated group and the control group in terms of the densities of CD68, IL-8, TNF-α, MCP-1, IL-6, and IL-1β (Figure 3A–F). The relative amount of CD68 cytokine of the treated group (23.2 ± 1.2) was around 30% less than that of the control group (33.9 ± 2.9) (Figure 3E). The IL-8 cytokine amount in the hydrocolloid-treated group (21.6 ± 0.8) was about 50% less than that of the control group (44.6 ± 2.3) (Figure 3F). Similar to the CD68 cytokine, the rest of the IF staining data show that the densities of TNF-α, MCP-1, IL-6, and IL-1β of the treated group are significantly smaller than those of the control group (Figure 3A–D).

On day 10 following the injury, double immunofluorescence staining reveals fewer numbers of iNOS of M1 in the wound region of the hydrocolloid-treated group (39.5 ± 3.4) than those in the control group (27.3 ± 1.4) (Figure 2G). Similarly, the intensity of F4/80 in the treated group (13.5 ± 0.7) is lower than that of the control group (22.2 ± 2.7) (Figure 2I).

LPS is a well-known pro-inflammatory agent in cell culture that is naturally produced by gram-negative bacteria. Figure 4 depicts the release of inflammatory mediators. The release of IL-6 and TNF-α was significantly affected by LPS treatment in different cell culture conditions (*p* = 0.05). Figure 4A shows the concentrations of IL-6 in cell fluids after 24 h of incubation. While the positive control (LPS) has the highest concentration of IL-6 (11.2 ± 1.1), the other two groups have IL-6 concentrations similar to that of the control group (5.9 ± 1.0) (HC + LPS: 6.6 ± 0.5 and HC: 6.4 ± 0.9). This demonstrates that HC could destroy substances that cause inflammation.

Additionally, we used TNF-α as a biomarker to test the pro-inflammatory capacity of HC in cell culture (Figure 4B). The result shows HC accelerates the pro-inflammatory response because the concentrations of TNF-α of the control group, the HC group, and the HC + LPS group are similar (0.144 ± 0.032, 0.133 ± 0.009, and 0.157 ± 0.014, respectively). Moreover, the highest concentration of TNF-α (1.991 ± 0.404) belongs to cell fluid containing only LPS (positive control).

### 3.3. Observation on the Proliferation of the Wound Healing Process

We used an immunofluorescence staining approach to examine the expression of α-SMA to investigate the role of a hydrocolloid in regulating myofibroblast activity in wound healing. As shown in Figure 2G, there was a significant increase in α-SMA expression on day 10 after treatment (16.7 ± 0.7 for the control and 27.1 ± 1.7 for treated rats; *p* < 0.0001). In addition, the result of the Western blot analysis demonstrated the dramatic rise in the α-SMA intensity between the control (0.355 ± 0.062) and the treated group (0.852 ± 0.115) (Figure 5A,B).

The cytoskeletal protein vimentin is released into the extracellular space after an injury, where it attaches to the cell surface of mesenchymal leader cells at the wound edge in the native matrix environment and aids wound closure [59]. The higher vimentin expression of the hydrocolloid group compared to that of the control group is shown by the immunofluorescence staining result (26.3 ± 0.8 and 35.1 ± 1.5, respectively; *p* < 0.0001) (Figure 2H) as well as the Western blot result (0.529 ± 0.088 and 1.080 ± 0.077, respectively; *p* = 0.0015) (Figure 5A,C). 

TGF-β3 has been found to generate α-smooth muscle actin (α-SMA) as a myofibroblast differentiation marker, which is critical for wound contraction [60]. Moreover, in wound healing, TGF-3 may play an anti-fibrotic role [61]. The Western blot result also showed that the TGF-β3 level in the hydrocolloid patch group was significantly higher than that level in the control group on day 10 (1.104 ± 0.147 and 0.635 ± 0.083, respectively) (Figure 5D).

Collagen helps wounds heal by attracting fibroblasts and encouraging the formation of new collagen in the wound bed. Hence, collagen formation in the healing process is one of the most important factors to determine that wounds heal quickly under product activation. While in the Western blot data of collagen III, the intensity of the control group (0.656 ± 0.060-fold) makes up 60% of the hydrocolloid group (1.077 ± 0.082-fold) (Figure 5E), the expression of collagen I showed that there is no statistical significance between the two groups (Figure 5F) on day 10.

M2 are anti-inflammatory macrophages that help resolve inflammatory processes by producing anti-inflammatory cytokines, which aid in tissue healing [62]. Our results indicated that Arg-1 expression of the treated group (29.9 ± 2.1) is slightly higher than that of the control group (22.8 ± 1.2) (Figure 5G,H). CD163 in M2 binds to hemoglobin–haptoglobin complexes, and they release anti-inflammatory mediators [63]. Double immunofluorescence staining showed that rats with HC had significantly higher densities of CD163 (31.4 ± 1.8) compared with control individuals (20.8 ± 0.8) (Figure 5G,I).

## 4. Discussion

The wound healing process includes four continuous stages: hemostasis, inflammation, proliferation, and maturation. As the maturation phase might take years to complete, in the present study, we focused on analyzing the first three stages by going from macroscopic and microscopic to molecular levels of information.

### 4.1. ZnO-NPs-HC Promotes Wound Reconstruction on Both Macroscopic and Microscopic Scales

A macroscopic image (Figure 1A) shows that the hemostasis in the ZnO-NPs-HC-treated group and the control group finished between day 0 and day 3. On day 7, there was a significant difference between the wound areas of the treated group and the control group. On day 10, the skin regeneration in the ZnO-NPs-HC-treated group was almost completed, while the healing of the control group was still in progress (Figure 1B), suggesting that our hydrocolloid product enhanced wound healing in our in vivo model on the macroscopic scale. In addition to the morphology observation, the microscopic observation also shows that the ZnO-NPs-HC-treated group had a faster healing speed than the control group regarding the thickness of the epidermis, granulation, and the collagen density. Both macroscopic information and microscopic information confirm that the ZnO-NPs-HC accelerated the healing process in the rat model.

### 4.2. ZnO-NPs-HC-Stimulated Inflammatory Phase Progression and Decrease in the Inflammatory Responses of Wound Healing

Nitric oxide regulates three essential aspects of the wound healing process, vascular homeostasis, inflammation, and antimicrobial action, particularly excessive NO associated with infected or highly inflamed wounds leading to tissue damage [53]. The level of NO in the treated group was significantly lower than that in the control group, indicating that the ZnO-NPs-HC had an impact on the inflammation phase of the wound healing.

In addition, reactive oxygen species (ROS) have an important role in orchestrating the normal wound healing process [64]. Excessive ROS generation triggers the activation of pro-apoptotic proteins, which leads to cell death and, in extreme situations, cellular necrosis [65]. In our study, there was no significant difference between the ROS levels of the ZnO-NPs-HC-treated group and the control group. This suggests that our product reduces ROS production (shown by the DCF-DA assay; Figure 2B) to accelerate the pro-inflammatory process.

Pro-inflammatory cytokines boost the proliferation and synthesis of antimicrobial peptides by keratinocytes, which helps heal acute wounds. However, overproduction of pro-inflammatory cytokines can prolong inflammation and wound healing. Accordingly, our data (immunofluorescence staining and ELISA) showed that the group with wounds treated with ZnO-NPs-HCs had much lower levels of pro-inflammatory cytokines than the control group. 

The immunofluorescence staining data showed that ZnO-NPs-HCs can reduce the levels of pro-inflammatory cytokines (CD68, IL-8, TNF-α, MCP-1, IL-6, and IL-1β) in the treated group by up to 50% compared to the control group (Figure 3A–F). Moreover, ELISA data showed that the pro-inflammatory cytokine expression (TNF-α, IL-6, MCP-1, and IL-8) in the ZnO-NPs-HC group was about half that of the control group (Figure 2C–F).

Macrophages are involved in three stages of adult wound healing: inflammation, proliferation, and remodeling. The local macrophage population shifts from predominately pro-inflammatory (M1-like phenotypes) to anti-inflammatory (M2-like phenotypes) when wounds heal [66]. On day 10 following injury, double immunofluorescence staining revealed fewer numbers of M1 in the wound region of the hydrocolloid-treated group than that of the control group.

### 4.3. ZnO-NPs-HC Encourages the Proliferation Phase

The result of wound healing is determined by the proliferation of fibroblasts. During wound healing, the activated fibroblast cell, which is one of the most important components of the neo-formed connective tissue, is regulated and transformed into the myofibroblast phenotype, which is defined by the neo-expression of α-SMA. We used an immunofluorescence staining approach to examine the expression of α-SMA to investigate the role of hydrocolloids in regulating myofibroblast activity in wound healing. Our data showed that the ZnO-NPs-HC group had significantly higher levels of α-SMA than the control group on day 10 (Figure 3G). 

A Western blot analysis also demonstrated the dramatic rise in the α-SMA intensity in the treated group compared to that in the control group (Figure 5A,B). The cytoskeletal protein vimentin is released into the extracellular space after an injury, where it attaches to the cell surface of mesenchymal leader cells at the wound edge in the native matrix environment and aids wound closure [59]. 

The higher vimentin expression of the ZnO-NPs-HC-treated group compared to that of the control group was supported by immunofluorescence staining (Figure 3H) and Western blot results (Figure 5A,C). Additionally, TGF-β3 levels strongly increased in the wounds of the group treated with ZnO-NPs-HCs compared to control (Figure 5D), indicating that the generation of α-SMA for wound contraction in the treated group was much stronger than that in the control group.

Collagen III formation in the ZnO-NPs-HC group was 40% higher than that in the control group, while collagen I showed no statistical significance between the two groups (Figure 5E,F) on day 10. Other studies have shown that collagen III is the first collagen to form during wound healing, although it is quickly replaced by collagen I, the most common skin collagen [67].

M2 macrophages are anti-inflammatory cells that produce anti-inflammatory cytokines to aid tissue healing [62]. Campbell et al., showed Arg-1 inhibition or depletion does not affect the amount of alternatively activated macrophages but is linked to enhanced inflammation, including increased influx of iNOS+ cells and matrix deposition abnormalities [68]. 

However, in our research, Arg-1 expression in the treated group increased slightly compared to that in the control group (Figure 5G,H). CD163 in M2 binds to hemoglobin–haptoglobin complexes and releases anti-inflammatory mediators [63]. Rats with ZnO-NPs-HCs had significantly higher densities of CD163 compared to the control group (Figure 5G,I), suggesting that ZnO-NPs-HCs promote tissue repair through the M2 mechanism. Several studies have investigated the application of NPs (e.g., gold and silver) in improving the wound healing rate [25,69,70,71,72,73,74]. 

In the work of Kim and colleagues, a combination of HC and Au-NPs macroscopically showed improved wound healing in rat models on day 15 [73]. They also used immunohistology staining to show that only the epidermis thickness in the treated group was higher than that in the control group. They analyzed connective tissue formation (collagen and MMP-1) and showed that Au-NPs-HC induces the regeneration and re-epithelization of wound skin by inhibiting oxidative stress. 

Compared to their work, our study provides additional histology data on collagen and granulation thickness and pro-inflammatory antibodies (CD68, IL-8, TNF-α, MCP-1, IL-6, IL-1β, iNOS, and F4/80) to confirm the efficacy of the product in the inflammation phase. For the proliferation phase, Kim and colleagues focused on analyzing the formation of connective tissue (collagen and MMP-1), while our study focused on myofibroblast activity (α-SMA, TGF-β3, vimentin), along with connective tissue (collagens I and III). 

Lau and colleagues also found that Au-NPs accelerated wound healing in rat models on day 9 [70]. However, the study was limited to only the microscopic level of information to confirm the effective healing of Au-NPs-HCs. Kim et al. proved that AgNPs do not delay epithelial wound healing in rabbit models [74] but AgNPs do not improve the wound healing rate either. Compared to previous studies that have combined hydrocolloid with NPs for wound treatment, our study shows that ZnO-NPs provide effective support for wound healing in both inflammation and proliferation stages. ZnO-NPs are cheaper than Au-NPs. Additionally, ZnO-NPs absorb UV radiation better than Au-NPs [75,76]. Therefore, our ZnO-NPs-HCs could be a promising product for wound treatment.

## 5. Conclusions

In this study, using Sprague–Dawley rat models, we showed that ZnO-NPs-HCs significantly improved wound healing after ten days in terms of the healing rate and stimulated inflammation and proliferation phases. We used a comprehensive set of biomarkers of proinflammatory cytokines (CD68, IL-8, TNF-α, MCP-1, IL-6, IL-1β, and M1) to confirm that ZnO-NPs-HCs effectively improved healing during the inflammation phase. This in vivo result was also supported by the results from in vitro models (RAW264.7 cells) through the expression of TNF-α and IL-6. We also confirmed that ZnO-NPs-HCs encouraged the proliferation phase of the healing process, shown by the expression of fibroblast biomarkers (α-SMA, TGF-β3, vimentin, collagen, and M2). 

Our results include a comprehensive analysis of wound healing by measuring the biomarker in each phase and suggest a cheaper method for wound dressing where expensive NPs (Au-NPs) are replaced by ZnO-NPs. Although in this study, ZnO-NPs-HCs patch improved the wound healing process, ZnO-NPs still have potential cytotoxicity [77]. Previous research suggested that combinations of ZnO-NPs and biocompatible materials, such as cellulose and chitosan, to minimize the cytotoxic effects of ZnO-NPs [78]. Future studies might be conducted to improve the current hydrocolloid-based wound dressing materials.

## Figures and Tables

**Figure 1 polymers-14-00919-f001:**
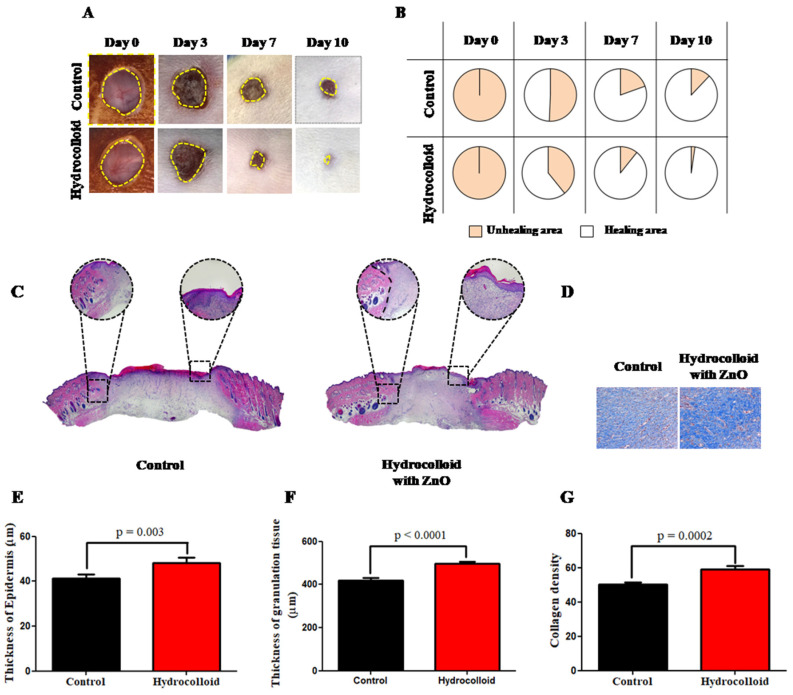
A hydrocolloid patch covered with ZnO-NPs accelerates wound healing on both macroscopic and microscopic scales. (**A**) Representative images (1× magnification) of the wound of the hydrocolloid-with-ZnO-NPs patch and the control group throughout the healing process. (**B**) Wound healing closure rates of these two groups. The rates presented as a percentage of the initial wound area on day 0. (**C**) 4× magnification images of H&E staining on day 10 after treatment with a hydrocolloid patch covered with ZnO-NPs and without treatment. (**D**) 10× magnification images of MT staining on day 10. The blue color indicates the distribution of the collagen. (**E**) Quantified epidermis thickness gap on day 10. (**F**) Quantified granulation tissue gap on day 10. (**G**) Quantified collagen density on day 10.

**Figure 2 polymers-14-00919-f002:**
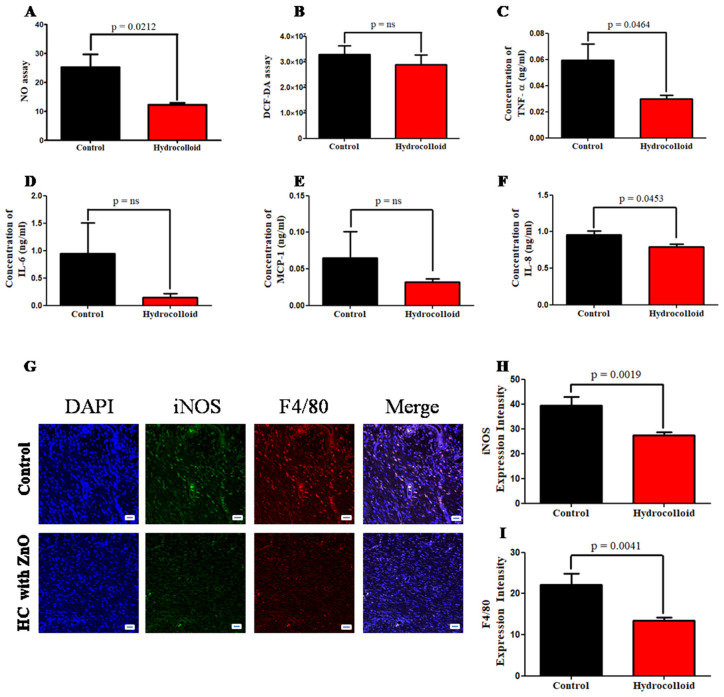
A hydrocolloid patch covered with ZnO nanoparticles accelerates the inflammatory phase progression and decreases the inflammatory responses of wound healing. (**A**) Nitrate assay. (**B**) DCF-DA assay. The concentration gap of pro-inflammation responses on day 10 was confirmed by ELISA between two groups. (**C**) Tumor necrosis factor-alpha, TNF-α. (**D**) Interleukin 6, IL-6. (**E**) Monocyte chemoattractant protein-1, MCP-1. (**F**) Interleukin 8, IL-8. (**G**) Fluorescent micrographs showing cytokine staining of macrophage 1 on day 10 of the wound healing process. The quantitative density gap of antibodies of macrophage 1 of the two groups on day 10. (**H**) Inducible nitric oxide synthase, iNOS. (**I**) F4/80. Scale bars are 50 µm.

**Figure 3 polymers-14-00919-f003:**
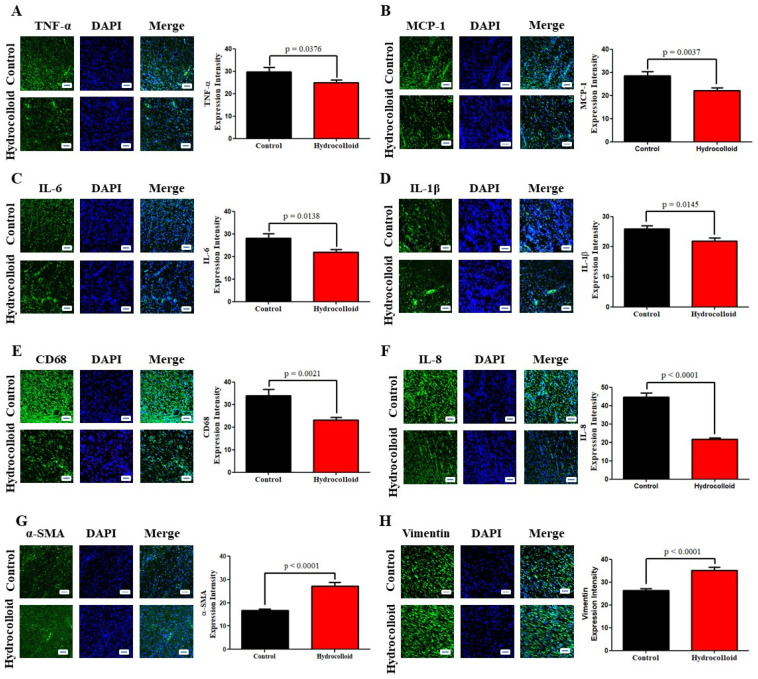
A hydrocolloid patch covered with ZnO nanoparticles bolsters inflammation and proliferative phases, which is confirmed by immunofluorescence staining. Fluorescent micrographs showing different cytokine staining of tissues growing on the posterior wound after 10 days and the quantitative density gap of the inflammation response of the two groups on day 10. (**A**) Tumor necrosis factor-alpha, TNF-α. (**B**) Interleukin 8, IL-8. (**C**) Interleukin 6, IL-6. (**D**) Interleukin 1 beta, IL-1β. (**E**) Cluster of Differentiation 68, CD68. (**F**) Monocyte chemoattractant protein-1, MCP-1. The quantitative density gap of proliferative response of the two groups on day 10. (**G**) Alpha smooth muscle actin, α-SMA. (**H**) Vimentin. Scale bars are 50 µm.

**Figure 4 polymers-14-00919-f004:**
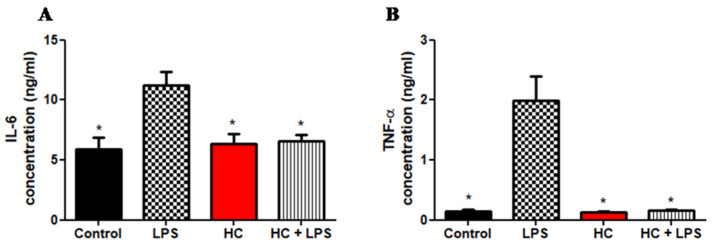
A hydrocolloid patch covered with ZnO nanoparticles accelerates the pro-inflammatory property of the RAW 246.7 cell culture. The concentration gap of pro-inflammation responses between four groups (control, LPS, HC, and HC + LPS) was confirmed by ELISA. (**A**) Interleukin 6, IL-6. (**B**) Tumor necrosis factor-alpha, TNF-α. The asterisks indicates *p* < 0.05.

**Figure 5 polymers-14-00919-f005:**
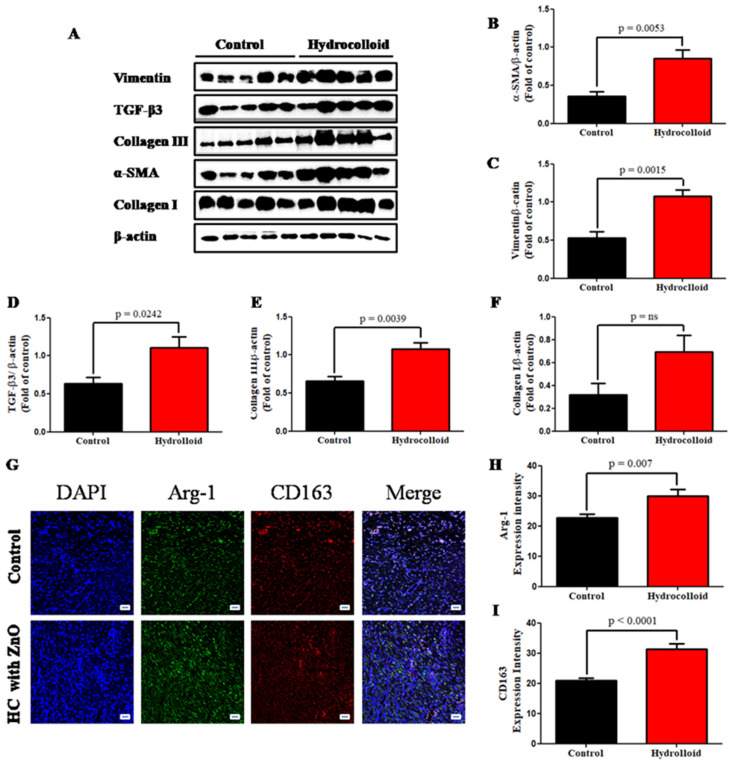
A hydrocolloid patch covered with ZnO nanoparticles accelerates proliferative phases and increases the proliferative responses of wound healing. (**A**) Representative images of Western blotting. Quantitative densitometry analysis of proliferative response expression. (**B**) Alpha smooth muscle actin, α-SMA. (**C**) Vimentin. (**D**) Transforming growing factor-beta 3, TGF-β3. (**E**) Collagen III. (**F**) Collagen I. (**G**) Fluorescent micrographs showing cytokine staining of macrophage 2 on day 10 of the wound healing process. The quantitative density gap of macrophage 1 antigens of the two groups on day 10. (**H**) Arginase 1, Arg-1. (**I**) CD163. Scale bars are 50 µm.

## Data Availability

Not applicable.

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
