# Peer review of "Evaluation of the Performance of a ZnO-Nanoparticle-Coated Hydrocolloid Patch in Wound Healing"

_polymers, 2022, doi:10.3390/polym14050919_

Round 1

Reviewer 1 Report

Dear authors

Your article's topics are interesting to this journal, the abstract and introduction are clear and concise with all the elements well structured, the methodology is well designed, and described just a couple of comments are below. In my point of view the results are logical and well presented and are properly discussed and cited

Comments and suggestions

a) Please use italics for all Latin expressions such as in vivo and in vitro

b) Please in methodology start with a material section, describing the materials and solvents used to prepare the hydrocolloid, and also add how the system was prepared, if you don't prepare this hydrocolloid, at least put that was donated by someone and reference the respective article, where it was synthesized and physicochemically characterized

c) Please use expressions in past and third person (revise line 108)

d) Please number the equation of line 118

e) For all micrographs add the scale bar (with the length described) and magnification of the pictures

Author Response

Responses to comments of the reviewer 1:

Point 1: “Please use italics for all Latin expressions such as in vivo and in vitro

Response to the point 1:

Thank you for your suggestion. We added italic format for Latin expressions based on your comment.

Point 2: “Please in methodology start with a material section, describing the materials and solvents used to prepare the hydrocolloid, and also add how the system was prepared, if you don't prepare this hydrocolloid, at least put that was donated by someone and reference the respective article, where it was synthesized and physicochemically characterized”

Response to the point 2:

Thank you for comments. Based on your suggestion, we added the description for preparation of hydrocolloid in the methodology as below:

In the revised manuscript (Page 2, Line 98-99):

The ZnO-NPs-HCs patch was kindly provided by CGBio company (South Korea). The patch was prepared by following protocols in a pending patent.

Point 3: “Please use expressions in past and third person (revise line 108).

Response to the point 3:

Thank you for your suggestion. We changed expressions to past and third person as below:

In the original manuscript (Page 3, Line 108-110):

After 10 days of treatment, the rats were asphyxiated with CO2, and we collected from each rat a skin sample that included the 12mm diameter circular hole with full-thickness skin around the wound

In the revised manuscript (Page 3 Line 121-123):

After 10 days of treatment, the rats were asphyxiated with CO2, and 12mm diameter circular hole with full-thickness skin around the wound samples were collected from each rat.

Point 4: “Please number the equation of line 118.

Response to the point 4:

Thank you for your suggestion. We numbered that equation as equation (1) in the revised manuscript.

Point 5: “For all micrographs add the scale bar (with the length described) and magnification of the pictures.

Response to the point 5:

Thank you for your suggestion. We added scale bars and magnification of the pictures in the revised manuscript.

Reviewer 2 Report

  • Some important data from this study must be mentioned in the abstract.
  • The materials section is missing.
  • All the methodologies need references.
  • Line 125: what is FBS/PBS? Write full form first. All the abbreviations for the first time must come in full form.
  • What is BSA in line 127? Line 131: DAPI? The full form is required first. Line 154: what is TBST?
  • Line 149: [ref]. remove it.
  • References 8, 18 are too old. Replace them with newly published papers. As well as refs 31-35.
  • It can be a good idea if the authors add future prospects at the end of the manuscript.
  • It is suggested to use the following references in this manuscript:

Sabbagh, F., Kiarostami, K., Mahmoudi Khatir, N., Rezania, S., & Muhamad, I. I. (2020). Green synthesis of Mg0. 99 Zn0. 01O nanoparticles for the fabrication of κ-Carrageenan/NaCMC hydrogel in order to deliver catechin. Polymers12(4), 861.

Sabbagh, F., Kiarostami, K., Khatir, N. M., Rezania, S., Muhamad, I. I., & Hosseini, F. (2021). Effect of zinc content on structural, functional, morphological, and thermal properties of kappa-carrageenan/NaCMC nanocomposites. Polymer Testing93, 106922.

Author Response

Responses to comments of the reviewer 2:

Point 1: “Some important data from this study must be mentioned in the abstract.”

Response to the point 1:

Thank you for your comment. Based on your suggestion, we added important data of this study to the abstract as below:

In the revised manuscript (Page 1, Line 20-29):

“Using Sprague Dawley rat models, we showed that during 10-day treatment a hydrocolloid patch covered with ZnO-NPs (ZnO-NPs-HC) macroscopically and microscopically stimulated the wound healing rate and improved wound healing in the inflammation phase, shown by reducing of proinflammatory cytokines (CD68, IL-8, TNF-α, MCP-1, IL-6, IL-1β, and M1) up to 50%. Results from the in vitro models (RAW264.7 cells) also supported these in vivo results, that ZnO-NPs-HCs improved wound healing in the inflammation phase by expressing similar level of proinflammatory mediators (TNF-α and IL-6) as negative control group. Moreover, ZnO-NPs-HCs also encourage the proliferation phase of the healing process, which was displayed by increasing expression of fibroblast biomarkers (α-SMA, TGF-β3, vimentin, collagen, and M2) up to 60%.”

Point 2: “The materials section is missing.”

Response to the point 2:

Thank you for your comment. Based on your suggestion, we added the material section as below:

In the revised manuscript (Page 2-3, Line 98-105):

“The ZnO-NPs-HCs patch was kindly provided by CGBio company (South Korea). The patch was prepared by following protocols in a pending patent.

Bovine Serum Albumin (BSA) and 10x Tris-Buffered Saline (TBS) were purchased from Biosesang company, South Korea. Tween 20 (extra pure grade) was purchased from Duksan company, South Korea. Fetal Bovine Serum (FBS) was purchased from Thermo Fisher, USA. Phosphate-buffered saline (PBS) was purchased from iNtRON Biotechnology company, South Korea. Tris buffered saline with Tween (TBST) solution was made from TBS and Tween 20.”

Point 3: “All the methodologies need references.”

Response to the point 3:

Thank you for your comment. Based on your suggestion, we added references for all methodologies as below:

In the revised manuscript (Page 3, Line 127-129):

“The second part was used for biochemistry analysis, such as Western blot [50], Enzyme-Linked Immunosorbent (ELISA) [51], DCF-DA [52], and nitrate assays [53].”

Point 4: “Line 125: what is FBS/PBS? Write full form first. All the abbreviations for the first time must come in full form.”

Response to the point 4:

Thank you for your comment. We added full form of chemicals before abbreviations in the revised manuscript.

In the revised manuscript (Page 3, Line 102-104):

“Fetal Bovine Serum (FBS) was purchased from Thermo Fisher, USA. Phosphate-buffered saline (PBS) was purchased from iNtRON Biotechnology company, South Korea.”

In the revised manuscript (Page 3, Line 139-141):

“After that, for the heat antigen retrieval (unmasking) step, an antigen retrieval buffer (pH 6.0) containing 10% FBS in PBS”

In the revised manuscript (Page 4, Line 171-173):

“Next, the remaining protein-binding sites in the coated wells were blocked in 2 h by adding 200 µL of blocking buffer (10% FBS in 1×PBS) to each well.”

Point 5: “What is BSA in line 127? Line 131: DAPI? The full form is required first. Line 154: what is TBST?”

Response to the point 5:

Thank you for your comment. We added full form of chemicals before abbreviations in the revised manuscript.

In the revised manuscript (Page 2, Line 100-101):

“Bovine Serum Albumin (BSA) and 10×Tris-Buffered Saline (TBS) were purchased from Biosesang company, South Korea.”

In the revised manuscript (Page 4, Line 146-148):

“Finally, all tissues were mounted with 4′,6-diamidino-2-phenylindole (DAPI VECTASHIELD®) (Vector Laboratories, USA) for fluorescence imaging and kept at -20°C”

In the revised manuscript (Page 3, Line 104-105):

“Tris buffered saline with Tween (TBST) solution was made from TBS and Tween 20.”

Point 6: “Line 149: [ref]. remove it.”

Response to the point 6:

Thank you for your comment. We deleted “[ref]” as you suggested.

Point 7: “References 8, 18 are too old. Replace them with newly published papers. As well as refs 31-35.”

Response to the point 7:

Thank you for your comment. Based on your suggestion, we replaced references 8, 18, 31-35 with new references to the revised manuscript.

Point 8: “It can be a good idea if the authors add future prospects at the end of the manuscript.”

Response to the point 8:

Thank you for your comment. Based on your suggestion, we added future prospects at the end of the manuscript as below:

In the revised manuscript (Page 13, Line 465-469):

“Although in this study, ZnO-NPs-HCs patch improved wound healing process, ZnO-NPs still have potential cytotoxicity [77]. Previous research suggested combination of ZnO-NPs and biocompatible materials such as cellulose and chitosan to minimize cytotoxic effects of ZnO-NPs [78]. Future studies might be conducted for further improving the current hydrocolloid-based wound dressing materials.”

Point 9: “It is suggested to use the following references in this manuscript:

Sabbagh, F., Kiarostami, K., Mahmoudi Khatir, N., Rezania, S., & Muhamad, I. I. (2020). Green synthesis of Mg0.99 Zn0.01O nanoparticles for the fabrication of κ-Carrageenan/NaCMC hydrogel in order to deliver catechin. Polymers12(4), 861.

Sabbagh, F., Kiarostami, K., Khatir, N. M., Rezania, S., Muhamad, I. I., & Hosseini, F. (2021). Effect of zinc content on structural, functional, morphological, and thermal properties of kappa-carrageenan/NaCMC nanocomposites. Polymer Testing93, 106922.”

Response to the point 9:

Thank you for your comment. Based on your suggestion, we added new references (Sabbagh et al., 2020 and Sabbagh et al., 2021) to the revised manuscript as below:

In the revised manuscript (Page 2, Line 74-79):

“Furthermore, zinc oxide nanoparticles (ZnO-NPs) have been used as an important ingredient in wound dressing products due to their ability to protect against UV light, their antibacterial properties, their capacity to increase swelling ratio of materials, and their interaction with cells that promote recovery from injury by accelerating cellular metabolism, DNA repair, cytokine regeneration, etc. [40–48].”
